# The Impact of Rotational Malalignment Following Intramedullary Nailing for Intertrochanteric Fractures on Patients’ Functional Outcomes: A Prospective Study

**DOI:** 10.3390/jfmk9040247

**Published:** 2024-11-24

**Authors:** Michail Vavourakis, Athanasios Galanis, Dimitrios Zachariou, Evangelos Sakellariou, Christos Patilas, Panagiotis Karampinas, Angelos Kaspiris, Meletis Rozis, John Vlamis, Elias Vasiliadis, Spiros Pneumaticos

**Affiliations:** 3rd Department of Orthopaedic Surgery, National & Kapodistrian University of Athens, KAT General Hospital, 14561 Athens, Greece; athanasiosgalanis@yahoo.com (A.G.); dimitriszaxariou@yahoo.com (D.Z.); vagossak@hotmail.com (E.S.); chris.pat90@gmail.com (C.P.); karapana@yahoo.com (P.K.); angkaspiris@hotmail.com (A.K.); mrozhs@gmail.com (M.R.); jvlamis@email.com (J.V.); eliasvasiliadis@yahoo.gr (E.V.); spirospneumaticos@gmail.com (S.P.)

**Keywords:** hip, hip anteversion, femur, femoral anteversion, rotational alignment, rotational deformity, functional outcome, mortality, fracture union, prospective study

## Abstract

Background & Aims: Rotational malalignment after intramedullary nailing of intertrochanteric fractures is a relatively common complication that may have a crucial impact on both functional outcome and a fracture’s healing properties, ultimately affecting the patient’s postoperative morbidity and mortality. Methods: 74 patients treated with intramedullary nailing due to an intertrochanteric fracture underwent a postoperative computed tomography (CT) scan of the hips and knees. The femoral anteversion difference (D angle) between the operated and healthy hip was calculated using the technique described by Jeanmart’s method. The patients were divided into two groups according to their D angle values: a D angle less than 15° was considered an acceptable rotational alignment (group A), while a D angle equal to or more than 15° was considered a rotational deformity (group B). Postoperatively, the functional level of each patient was evaluated at the 6-month and 1-year follow-up visits and compared to the pre-fracture status using the modified Harris hip score (mHHS). Also, fracture union, other postoperative complications, and patient mortality were noted. Results: The mean femoral anteversion difference was 12.3° with a standard deviation of 10.3°. Of the 74 patients, 51 (68.9%) were assigned to group A and 23 (31.1%) to group B. At the 6-month postoperative follow-up, 67 (90.5%) patients had survived and 7 (9.5%) were deceased, with our statistical analysis indicating a linear trend (*p*-value = 0.048) between the presence of rotational malalignment and 6-month mortality. At the 1-year postoperative follow-up, 63 (85.1%) patients survived and 11 (14.9%) were deceased, with the statistical analysis indicating a significant relationship (*p*-value = 0.031) between the presence of rotational malalignment and the 1-year mortality. Regarding the functional outcome six months after the operation, the difference between the pre-fracture and the postoperative mHHS was 8.7/100 with a standard deviation of 6.1 for the 49 patients in group A and 14.5/100 with a standard deviation of 12.4 for the 18 patients in group B, with the statistical analysis indicating a significant difference (*t* = −2.536, significance < 0.05) in the functional level between the two groups. As for the functional outcome one year after the operation, the difference between the pre-fracture and the postoperative mHHS was 4.9/100 with a standard deviation of 7.8 for the 47 patients in group A and 8.3/100 with a standard deviation of 13 for the 16 patients in group B, with the statistical analysis indicating no significant difference (*t* = −1.266, significance > 0.05) in the functional level between the two groups. The only postoperative complication noted was fracture non-union, presenting in two patients (3%), with the statistical analysis indicating no significant relationship (*p*-value = 0.698) between this complication and the presence of rotational malalignment. Conclusions: In this study, rotational malalignment after intertrochanteric fracture intramedullary nailing presents in 31.1% of cases. The correlation between this malalignment and functional outcomes shows a decline in patients’ functional abilities during the first six postoperative months, a condition that improves over time, with most patients returning to their preoperative functional status one year after the operation. Furthermore, our results indicate a possible relationship between rotational malalignment and mortality within one year. Future research should focus on creating a more detailed, functional evaluation system for the elderly and applying it to a larger sample to confirm these findings.

## 1. Introduction

Intertrochanteric femoral fractures are one of the most common causes of attendance at the orthopedic emergency department [1] and constitute a severe issue for the public healthcare system, as well as patients and their families. The cost of complete treatment for an intertrochanteric fracture is relatively high, as it includes not only the expenses of the operation and hospitalization but also the costs of rehabilitation in the hospital and at home [2]. Intertrochanteric fractures are usually the result of a fall from a standing height [3], mainly affecting older individuals. Approximately 90% of patients are over the age of 65, with an average age of 81 years [4,5]. After such an event, the patient will face significant difficulties in carrying out daily activities during the postoperative period, while approximately 75% of them will not be able to regain their previous functional status and independence [6,7]. Also, the increase in morbidity further exacerbates the disability and the deterioration in life quality. The mortality rate of these patients stands at 7–10% within the first month, reaches 20% six months after the injury, and rises to 36% within a year [8]. The usual treatment for these fractures involves closed reduction and internal fixation with the patient positioned on a traction table [9]. Assessing the reduction quality in the frontal and sagittal planes during surgery is usually straightforward, but doing the same in the transverse plane is more challenging, with rotational deformity being a relatively common postoperative finding [10]. Proper intraoperative fracture realignment is crucial, as malposition may affect both the patient’s functional outcome and the fracture’s healing properties. To our knowledge, this is the first study to evaluate, in detail, the impact of rotational malalignment after the intramedullary nailing of intertrochanteric fractures on patients’ functional outcomes and to investigate its possible correlation with patients’ morbidity and any other postoperative complications.

## 2. Materials and Methods

A prospective study on 74 patients suffering from an intertrochanteric femoral fracture was conducted in a tertiary trauma center from October 2021 to July 2023. Patients with a history of any previous operation on the hip, femur, or knee, those suffering from neurological conditions affecting their mobility, those with congenital deformities of the lower limbs, or patients who presented with accompanying injuries that made immediate postoperative mobilization impossible were excluded from this study.

All the patients were treated with intramedullary nail fixation (Figure 1) by the same surgical team. The operation was performed 24–48 h after the injury.

Each patient underwent a computed tomography (CT) scan of the hips and knees postoperatively after signing an informed consent and when they were pain free. The femoral anteversion of the operated and healthy hip, as well as the femoral anteversion difference between them (D angle), were calculated on the CT scan (Figure 2) using the method described by Jeanmart et al. in 1983 [11].

Measurements were performed twice by one surgeon and once by another one. Mean intraobserver and interobserver agreement was calculated as 1.6° and 1.9°, respectively. The patients were divided into two groups based on the measured anteversion difference. Patients with a D angle less than 15° were considered in the range of acceptable rotational alignment and were assigned to group A, while those presenting with a D angle equal to or more than 15° represented a rotational deformity and were assigned to group B.

Every patient was mobilized on the first postoperative day with partial weight-bearing using a walker. Follow-up visits were scheduled postoperatively at the 1st, 3rd, 6th and 12th months, where hip X-rays and a thorough physical examination were performed. Fracture union was evaluated at the 6-month X-ray, while the patient’s functional level was evaluated at the 6th and 12th month visits by using the modified Harris hip score (mHHS) validated in the Greek language. The scores can range from 0 to 100, with higher scores indicating better function and less pain. A score above 70 is often considered satisfactory, while scores below 60 suggest poor outcomes. Pre-fracture functional status was estimated based on each patient’s statements, using the mHHS to determine the difference in postoperative functional outcomes. Patient mortality and any other postoperative complications were noted.

Statistical analysis was done using Student’s *t*-test for the continuous variables and Pearson’s chi-square test for the categorical variables. SPSS (version 27.0) was used for the statistical calculations. Statistical significance was determined at a *p*-value < 0.05.

## 3. Results

A total of 74 patients were included in this study. The mean patient age was 80.6, ranging from 47 to 99 years. Most of them, namely 47 (63.5%), were females, while the rest, 27 (26.5%), were male patients. The most common cause of injury was a fall from a standing height, accounting for 68 cases (91.9%), with the remaining 6 cases (8.1%) resulting from a high-energy injury. In 40 cases, the affected hip was the left one, while the rest of the 34 cases (45.9%) suffered an injury to the right hip (Table 1). Time to operation did not affect patient outcomes. The same applies for the mechanism of injury, as a possible correlation between low- or high-energy injuries and postoperative differences in anteversion was not observed.

The mean value of femoral anteversion on the healthy hip was 13.3° with a standard deviation of 7.2°. In comparison, on the operated hip, the mean value of femoral anteversion was 23.3° with a standard deviation of 13.3°. The mean value of the femoral anteversion difference between the operated and healthy hip (D angle) was 12.3° with a standard deviation of 10.3° (Table 2).

Postoperatively, 51 patients (68.9%) presented with a femoral anteversion difference of less than 15°, while 23 patients (31.1%) presented with a femoral anteversion difference equal to or higher than 15° (Table 3).

Six months after the operation, out of the 74 patients enrolled, 67 (90.5%) survived and 7 (9.5%) were deceased. Specifically, out of the seven deceased patients, two (28.6%) belonged to group A, and five (71.4%) belonged to group B (Table 4). The statistical analysis results (Table 5) indicate a linear trend (*p*-value = 0.048) between the difference in femoral anteversion difference and the mortality within six months.

One year after the operation, out of the 74 patients enrolled, 63 (85.1%) survived and 11 (14.9%) were deceased. Specifically, four cases (36.7%) belonged to group A, and seven (63.3%) belonged to group B (Table 6). The statistical analysis (Table 7) results indicate a statistically significant relationship (*p*-value = 0.031) between the difference in femoral anteversion and mortality within one year.

The evaluation of patient functional level was based on the mHHS. Before the fracture, the mHHS ranged from 50.6 to 100, with a mean value of 86.7 and a standard deviation of 11.8. Six months after the operation, the 67 surviving patients presented with a mHHS ranging from 20.9 to 100, with a mean value of 77.9 and a standard deviation of 13.3. One year after the operation, the 63 surviving patients presented with a mHHS ranging from 20.9 to 100, with a mean value of 83.2 and a standard deviation of 14.3 (Table 8).

Table 9 depicts the correlation of the femoral anteversion difference between the operated and the healthy hip (D angle) to the functional outcome (pre-fracture to postoperative mHHS difference) six months after the operation. The 49 surviving patients in group A presented a mean mHHS difference value of 8.7/100 with a standard deviation of 6.1, and the 18 surviving patients in group B presented a mean mHHS difference value of 14.5/100 with a standard deviation of 12.4. The results of the statistical analysis indicate a statistically significant difference (*t* = −2.536, *p* < 0.05) in the pre-fracture to postoperative mHHS difference between the two groups (Table 9).

Table 10 depicts the correlation of the femoral anteversion difference between the operated and the healthy hip (D angle) to the functional outcome (pre-fracture to postoperative mHHS difference), one year after the operation. The 47 surviving patients in group A presented a mean mHHS difference value of 4.9/100 with a standard deviation of 7.8, and the 16 surviving patients in group B presented a mean mHHS difference value of 8.3/100 with a standard deviation of 13. The results of the statistical analysis indicate that there is no statistically significant difference (*t* = −1.266, *p* > 0.05) in the pre-fracture to postoperative mHHS difference between the two groups (Table 10).

Regarding fracture union, only 2 (3%) out of the 67 surviving patients presented with a non-union at the 6-month follow-up hip X-ray. One of them (50%) belonged to group A and the other one (50%) to group B (Table 11). The results of the statistical analysis (Table 12) indicate that there is no statistically significant relationship (*p*-value = 0.698) between the difference in femoral anteversion and fracture union six months after the operation, as the values did not exceed the thresholds of significance (*p*-value > 0.05).

## 4. Discussion

In this study, a rotational malalignment after the intramedullary nailing of intertrochanteric fractures was observed in 31.1% of the cases. Detecting rotational malalignment during the intraoperative reduction of an intertrochanteric fracture is challenging using a standard radiographic evaluation. This is due to the limited ability of this type of assessment to appraise the rotational alignment of the fracture reduction in the transverse plane, as it only allows for the evaluation of reduction quality in the frontal and sagittal planes [10]. There have been numerous studies proposing various methods to avoid torsional malalignment during the intramedullary nailing of diaphyseal and distal femoral fractures [12,13,14,15,16], but there are only a few, and they are not fully substantiated on the issue regarding intertrochanteric fractures [17,18,19]. Numerous methods have been outlined in terms of the measurement criteria on a CT scan. However, it has yet to be conclusively proven that any of those methods are better than the others, as they all exhibit similar standard deviation values and intraobserver–interobserver variability [20]. For this study, we opted to utilize the technique described by Jeanmart et al. in 1983 [11], aiming to present results comparable to the limited existing research on this specific issue [17,18,19].

Our sample was divided into two groups according to the difference in femoral anteversion between operated and healthy hips (D angle). A D angle value less than 15° was considered an acceptable rotational alignment, as the average femoral anteversion difference in the general population is estimated to range from 2.9° to 8.8° [21,22]. Those 51 patients (68.9%) were assigned to group A. Conversely, a D angle equal to or more than 15° was considered a rotational malalignment based on the findings of the study by Jaarsma et al. [23]. This definition is also consistent with the findings of other studies [22], which estimate the maximum femoral anteversion difference in the general population to range from 12° to 13°. Those 23 patients (31%) were assigned to group B. Compared to other studies on this specific issue, the incidence rate is lower than the 40% found by Ramanoudjame et al. [17], but slightly larger than the 25.7% found by Kim et al. [18] and the 24.3% found by Annappa et al. [19].

In the existing literature, there are only a few references regarding the impact of femoral shaft rotational malalignment on the functional outcome of patients following intramedullary nailing, but almost none regarding intertrochanteric fractures. According to Jaarsma et al. [23], this could be attributed to the lack of a reliable and universally accepted evaluation system to assess the influence of rotational deformity on functional outcomes. The only study [18] that attempted to evaluate the correlation of rotational malalignment following the intramedullary nailing of an intertrochanteric fracture to clinical outcomes utilized the Koval score and did not find a statistical significance (*p*-value = 0.458) for this relationship. The authors concluded that their clinical results may have been influenced by the potential inadequacy of the Koval score in assessing the functional level in older patients.

Based on this assumption, we utilized the modified Harris hip score to evaluate patients’ functional levels in the current study. The correlation between the femoral anteversion difference and the functional outcome revealed a significant relationship (*t* = −2.536, *p* < 0.05) six months after the operation, with patients who presented with a rotational deformity exhibiting a more significant difference between the pre-fracture and the postoperative mHHS, thus falling short of their pre-fracture functional level, compared to patients presenting with an acceptable rotational alignment. On the contrary, one year after the operation, the correlation between the femoral anteversion difference and the functional outcome did not reveal a significant relationship (*t* = −1.266, *p* > 0.05), with all the patients exhibiting similar pre-fracture to postoperative mHHS differences, thus approaching their pre-fracture functional levels, regardless of the observed rotational alignment. Table 13 compares our study’s results regarding functional outcomes with the results provided by Kim et al. [18].

Most studies analyzing the impact of rotational malalignment after intramedullary nailing on the functional outcome focus on cases of diaphyseal or distal femoral fractures and conclude that patients presenting with a femoral anteversion difference greater than 15° have more functional limitations and lower satisfaction levels [23,24]. Additionally, these studies agree that external rotational deformities usually have a greater impact and are less tolerated by patients than internal rotational deformities [25,26]. In order to compensate for the external rotation of the lower limb during walking, the femoral neck must be placed in retroversion, a position less tolerated due to pain caused by impingement [27,28].

Of course, these studies usually concern younger, higher-demand patients, for whom postoperative functional expectations are entirely different than in older individuals. According to our study’s results, rotational deformity following the intramedullary nailing of an intertrochanteric fracture seems to impact the patients’ functional level mainly during the first six postoperative months, as opposed to the second six-month postoperative period, during which most patients show an improvement, approaching their pre-fracture functional levels, regardless of the postoperative difference in the femoral anteversion. However, it should be noted that our sample mainly consists of older patients, with an average age of 80.6 years, so the results of a similar study in a sample of younger patients may reveal a different impact on functional levels.

In the current study, only a few patients experienced any postoperative complications. One of the possible complications that was meticulously studied was fracture non-union, as it was thought to be closely associated with rotational alignment, as well as having a significant impact on functional outcome. The evaluation of fracture union was based on radiographic imaging in combination with a clinical examination at the 6-month follow-up. Only two (3%) of our patients presented with a non-union (Figure 3) and were reoperated on. No relationship was found between the postoperative difference in femoral anteversion and fracture union, as the statistical values did not exceed the thresholds of significance (*p*-value = 0.698).

To our knowledge, this is the first study to address the possible effect of rotational malalignment after the intramedullary nailing of intertrochanteric fractures on patients’ mortality. Regarding the first six months after the operation, out of the seven deceased patients, five (71,4%) presented with a postoperative rotational deformity. The statistical analysis indicated a significant linear trend (*p*-value = 0.048) between the postoperative difference in femoral anteversion and mortality within six months. One year after the operation, 7 (63.6%) out of the 11 deceased patients presented with a postoperative rotational deformity, with the statistical analysis indicating a statistically significant relationship (*p*-value = 0.038) between the difference in femoral anteversion and the mortality within one year. To sum up, rotational deformity after intramedullary nailing has a significant impact on the mortality of patients suffering from an intertrochanteric fracture.

We are fully aware of the limitations present in this study. Since intertrochanteric fractures usually occur in older people, we focused on this population. The mean age of our sample was 80.6 years, ranging from 47 to 99 years. The impact of rotational malalignment on functional outcomes in older individuals may appear to be of low significance, as most patients reached their pre-fracture functional level after one year. However, intertrochanteric fractures can also occur in younger individuals following a high-energy injury, and the influence of such a complication may be significant. Moreover, the lack of a more precise evaluation system, particularly in older individuals, may affect our functional outcome results. Finally, in analyzing the relationship between rotational malalignment and mortality, we did not take into account the patient’s comorbidities, which might have had a substantial impact on their potential postoperative demise. The cause of death for the majority of patients was attributed to cardiovascular disease, mainly myocardial infarction. In the future, studies on a larger population sample using a more detailed functional system may potentially alter the significance of rotational alignment in older individuals and question the correlation with patient mortality.

Despite these limitations, this study holds significance, as it is the only one in the current literature correlating rotational malalignment after intertrochanteric fracture intramedullary nailing to patient mortality, and only the second one to examine the operation’s effect on functional outcomes and the incidence of other postoperative complications, such as non-union.

## 5. Conclusions

In this study, rotational malalignment after the intramedullary nailing of intertrochanteric fractures was observed in 31.1% of cases despite the proper use of radiographic imaging during intraoperative fracture reduction. Our findings suggest that rotational malalignment leads to a decline in patients’ functional abilities during the initial six-month postoperative period. This condition tends to improve by the end of the first postoperative year, with most patients almost returning to their preoperative functional status. Additionally, we observed that rotational malalignment may have a strong relationship with mortality in these patients, although this cannot be clearly stated due to the lack of a correlation with each patient’s comorbidities. Moving forward, it is crucial for upcoming research to develop a more accurate evaluation system to assess the functional capabilities of older individuals and to test this system on a larger population sample to further validate these findings.

## Figures and Tables

**Figure 1 jfmk-09-00247-f001:**
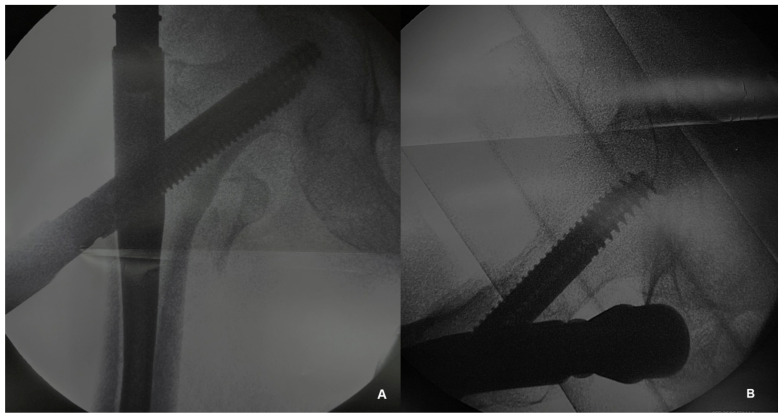
Intraoperative radiographic assessment during the intramedullary nailing of an intertrochanteric fracture. (**A**) Anteroposterior view and (**B**) lateral view.

**Figure 2 jfmk-09-00247-f002:**
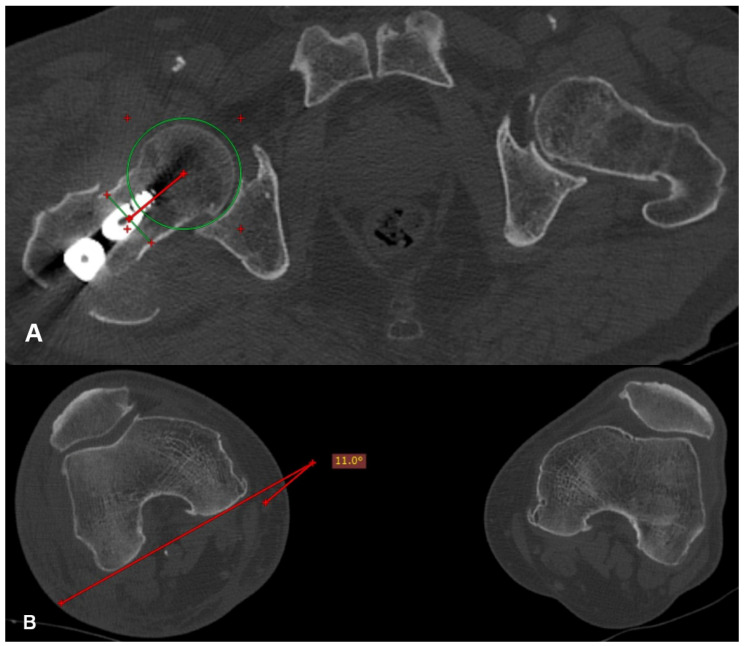
(**A**) A line is drawn that connects the center of the head, in the transverse section where it presents its maximum diameter, with the center of the neck, in the transverse section where it presents its narrowest width (redline). (**B**) Femoral anteversion is the projective angle formed between the tangent line at the posterior femoral condyles and the line connecting the center of the head with the center of the neck (red lines).

**Figure 3 jfmk-09-00247-f003:**
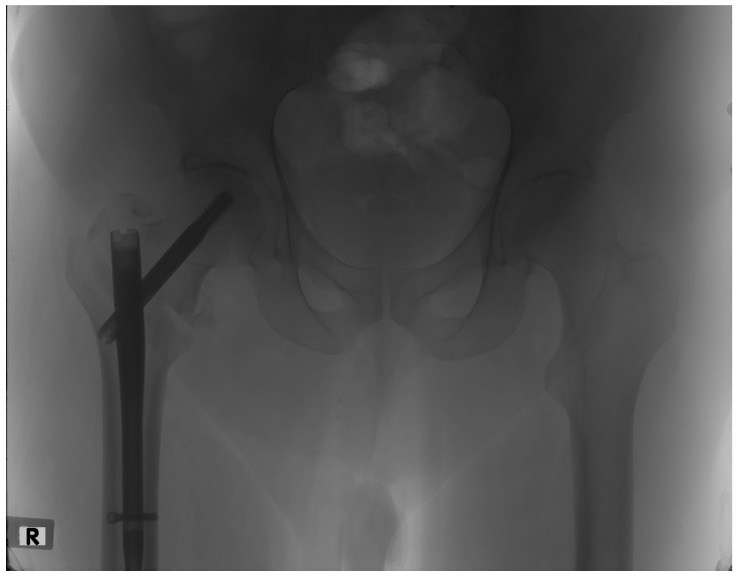
Hip X-ray of a patient presenting with an intertrochanteric fracture non-union at the 6-month follow-up.

**Table 1 jfmk-09-00247-t001:** Incidence and relative incidence of the sample’s characteristics.

		*N*	%
Gender	Male	27	36.5%
	Female	47	63.5%
Injury mechanism	Low-energy	68	91.9%
	High-energy	6	8.1%
Affected hip	Right	34	45.9%
	Left	40	54.1%

**Table 2 jfmk-09-00247-t002:** Analysis of femoral anteversion values.

Femoral Anteversion	Minimum Value	Mean Value	Standard Deviation	Range	Maximum Value
Healthy hip	2.2°	13.3°	7.2°	34.5°	36.7°
Operated hip	1.2°	23.2°	13.3°	61.2°	62.4°
Difference	0.9°	12.3°	10.3°	47.6°	48.5°

**Table 3 jfmk-09-00247-t003:** The incidence and relative incidence of the femoral anteversion difference (D angle).

D Angle	Group	*N*	%
<15°	A	51	68.9%
≥15°	B	23	31.1%

**Table 4 jfmk-09-00247-t004:** Frequency and relative frequency cross-table between the femoral anteversion difference (patient group) and mortality within six months.

			Group		Total
			A	B	
6-month mortality	Yes	*N*	2	5	7
		%	4%	21.7%	9.5%
	No	*N*	49	18	67
		%	96%	78.3%	90.5%
Total		*N*	51	23	74
		%	100%	100%	100%

**Table 5 jfmk-09-00247-t005:** Results of the independence testing for the variables difference in anteversion between the operated and healthy hip (patient group) and the mortality within six months.

	Value	Degrees of Freedom	Asymptotic Significance (2-Sided)
Pearson Chi-Square	6.072	2	0.048

**Table 6 jfmk-09-00247-t006:** Frequency and relative frequency cross-table of the femoral anteversion difference (patient group) and mortality within one year.

			Group		Total
			A	B	
1-year mortality	Yes	*N*	4	7	11
		%	7.8%	30.4%	14.9%
	No	*N*	47	16	63
		%	92.2%	69.6%	85.1%
Total		*N*	51	23	74
		%	100%	100%	100%

**Table 7 jfmk-09-00247-t007:** Results of the independence testing for the variable difference in anteversion between the operated and healthy hip (patient group) and the mortality within six months.

	Value	Degrees of Freedom	Asymptotic Significance (2-Sided)
Pearson Chi-Square	6.924	2	0.031

**Table 8 jfmk-09-00247-t008:** Analysis of patients’ pre-fracture and postoperative functional level based on the modified Harris hip score (mHHS).

	*N*	Minimum Value	Mean Value	Standard Deviation	Maximum Value
Pre-fracture mHHS	74	50.6	86.7	11.8	100
6-month postoperative mHHS	67	20.9	77.9	13.3	100
1-year postoperative mHHS	63	20.9	83.2	14.3	100

**Table 9 jfmk-09-00247-t009:** Cross-table between patient group (hip anteversion difference) and functional outcome (pre-fracture to postoperative functional level difference based on the modified Harris hip score) six months after the operation.

	Group (D Angle)	N of Surviving Patients	Mean Value	Standard Deviation	*t*	Degrees of Freedom	Significance (2-Tailed)
Difference between pre-fracture and six-month postoperative mHHS	A (<15°)	49	−8.7	6,1	−2.536	65	0.014
	B (≥15°)	18	−14.5	12,4	−1881	20.055	0.075

**Table 10 jfmk-09-00247-t010:** Cross-table between patient group (hip anteversion difference) and functional outcome (pre-fracture to postoperative functional level difference based on the modified Harris hip score) one year after the operation.

	Group (D Angle)	N of Surviving Patients	Mean Value	Standard Deviation	*t*	Degrees of Freedom	Significance (2-Tailed)
Difference between pre-fracture and one-year postoperative mHHS	A (<15°)	47	−4.9	7.8	−1.266	61	0.210
	B (≥15°)	16	−8.3	13.0	−0.995	18.820	0.332

**Table 11 jfmk-09-00247-t011:** Frequency and relative frequency cross-table between the femoral anteversion difference (patient group) and fracture union six months after the operation.

			Group		Total
			A	B	
Fracture union	Yes	*N*	48	17	65
		%	98%	94.4%	97%
	No	*N*	1	1	2
		%	2%	5.6%	3%
Total		*N*	49	18	67
		%	100%	100%	100%

**Table 12 jfmk-09-00247-t012:** Results of the independence testing for the variable difference in anteversion between the operated and healthy hip (patient group) and the fracture union six months after the operation.

	Value	Degrees of Freedom	Asymptotic Significance (2-Sided)
Pearson Chi-Square	0.720	2	0.698

**Table 13 jfmk-09-00247-t013:** Comparison of patients’ recovery to pre-fracture functional status between studies.

Study	Evaluation Score	Group (D Angle)	*N*	Pre-Fracture Functional Level Recovery	Significance (*p*-Value)
Kim et al. [18]	Koval	A (<15°)	81	61 (75.3%)	0.458
		B (≥15°)	28	23 (82.1%)	
Present study	Modified Harris Hip	A (<15°)	47	35 (74.5%)	0.210
		B (≥15°)	16	12 (75%)	0.332

## Data Availability

All raw data are available to access upon request.

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
