# Peer review of "The Impact of Rotational Malalignment Following Intramedullary Nailing for Intertrochanteric Fractures on Patients’ Functional Outcomes: A Prospective Study"

_jfmk, 2024, doi:10.3390/jfmk9040247_

Round 1

Reviewer 1 Report

Comments and Suggestions for Authors

Title Review: The title "The Impact of Rotational Malalignment Following Intramedullary Nailing for Intertrochanteric Fractures on Patients’ Functional Outcome: A Prospective Study" precisely describes the study. Congratulations!

Acronym Usage: Please remember to always include the acronym for CT when first introducing it in the abstract.

Group Naming: Why are 'Group A' and 'Group B' used? Wouldn't it be more intuitive to use 'Normal D Angle Group' and 'Abnormal D Angle Group' instead?

Patient Distribution: Considering that 23 out of 74 patients (31%) were assigned to Group B, was this distribution also expected based on the literature?

Introduction Clarity: Your introduction convincingly addresses the risks of postoperative rotational deformity. However, it is not clear what the novelty of the study is compared to existing research. What has already been done in this context?

Visual Aids: It would be very instructive for the reader if the authors added a figure showing, on one side, an image of a patient with a D angle of less than 15° (normal) and, on the other side, an image of a patient presenting with a D angle of 15° or greater (the most severe case).

Injury Analysis: In your study, the majority of the sample sustained injuries from a fall from standing height (91.9%). Would those who suffered high-energy injuries be more likely to be included in the group presenting with a D angle of 15° or greater? Additionally, why were only six individuals included who sustained high-energy impact injuries?

Statistical Organization: The statistical analysis could be improved. For instance, when discussing comparisons and correlations, the information in the text does not appear to be well organized. You state that 4 cases belonged to group A and 7 cases belonged to group B (Table 5). Looking at this result, it doesn't seem there is a significant statistical difference between 4 and 7, leading me to believe that mortality was not influenced by the patient's condition. If you are drawing conclusions solely from correlations, the presentation of the results should be enhanced, as it is not perfectly clear.

Result Descriptions: The most important improvement for the article is to enhance the description of the results from a statistical perspective. It is unclear what your intentions are regarding the mortality data, for example. You need to be more explicit and pragmatic. My suggestion is to display the p-values when making comparisons in the table and to include figures showing the correlations with both p-values and correlation coefficients (r) between the variables.

Harris Hip Score (mHHS): It would be beneficial for the lay reader to understand the Harris Hip Score (mHHS). A brief explanation of its interpretation would be helpful. For instance, scores range from 0 to 100, with higher scores indicating better function and less pain. A score above 70 is often considered satisfactory, while scores below 60 suggest poor outcomes. Providing this context would enhance the reader's comprehension.

Clarification in Tables:

  • In Table 7, please clarify that the numbers 49 and 18 represent the 67 surviving patients at the 6-month follow-up.
  • In Table 8, please clarify that the numbers 47 and 16 represent the 63 surviving patients at the 1-year follow-up.

Mean Value Interpretation: The mean values shown in Tables 7 and 8 represent the differences concerning the pre-fracture status, which is a good idea. However, it would be beneficial to include polarity indicators (positive or negative). For example, while Group B showed the greatest differences, it is unclear whether these represent increases or decreases compared to the pre-fracture status.

Conclusion Focus: Your conclusion has become somewhat unfocused. It could be more tightly aligned with the objectives of the study.

Video Resource: Would it be possible to include a link to a video that visually demonstrates some of the biomechanical movements related to the concepts discussed in the article, such as rotational malalignment, femoral anteversion, and external rotation of the lower limb? This would enhance the reader's understanding.

Author Response

Dear reviewer, 

Thank you for the valuable comments and the time spent reviewing our work.

Acronym Usage: Please remember to always include the acronym for CT when first introducing it in the abstract.

Answer: The issue was addressed both in the abstract and the main text.

Group Naming: Why are 'Group A' and 'Group B' used? Wouldn't it be more intuitive to use 'Normal D Angle Group' and 'Abnormal D Angle Group' instead?

Answer: The terms “group A” and “group B” were preferred due to their easier use (short description) in both the text and tables.

Patient Distribution: Considering that 23 out of 74 patients (31%) were assigned to Group B, was this distribution also expected based on the literature?

Answer: “Compared to the other studies on this specific issue, the incidence rate is lower than the 40% of Ramanoudjame et al. [17], but slightly larger than the 25.7% of Kim et al. [18] and the 24.3% by Annappa et al. [19].” (added to the second paragraph of the discussion section)

Introduction Clarity: Your introduction convincingly addresses the risks of postoperative rotational deformity. However, it is not clear what the novelty of the study is compared to existing research. What has already been done in this context?

Answer: “To our knowledge, this is the first study aiming to evaluate in detail the impact of the rotational malalignment after intramedullary nailing of intertrochanteric fractures on the patient’s functional outcome and to investigate its possible correlation with patients’ morbidity and any other postoperative complications.” The last paragraph in the introduction states that this the first study evaluating these in detail.

Visual Aids: It would be very instructive for the reader if the authors added a figure showing, on one side, an image of a patient with a D angle of less than 15° (normal) and, on the other side, an image of a patient presenting with a D angle of 15° or greater (the most severe case).

Answer: We were not able to attain an actual patient image depicting a severe malformation, even in our worst cases. Despite that, we could add a postoperative X-ray of such a patient if you find it more helpful.

Injury Analysis: In your study, the majority of the sample sustained injuries from a fall from standing height (91.9%). Would those who suffered high-energy injuries be more likely to be included in the group presenting with a D angle of 15° or greater? Additionally, why were only six individuals included who sustained high-energy impact injuries?

Answer: We didn’t aim to monitor fewer patients sustaining a high-energy injury, we assume that the small sample correlates to the sample’s age as these fractures typically occur from a low-energy mechanism in the elderly. As for a possible connection between the high-energy injuries and the group presenting with a D angle of 15° or greater, no correlation was observed (comment added on the end of the first paragraph on the results section).

Statistical Organization: The statistical analysis could be improved. For instance, when discussing comparisons and correlations, the information in the text does not appear to be well organized. You state that 4 cases belonged to group A and 7 cases belonged to group B (Table 5). Looking at this result, it doesn't seem there is a significant statistical difference between 4 and 7, leading me to believe that mortality was not influenced by the patient's condition. If you are drawing conclusions solely from correlations, the presentation of the results should be enhanced, as it is not perfectly clear.

Answer: Statistical descriptions and additional tables have been provided.

Result Descriptions: The most important improvement for the article is to enhance the description of the results from a statistical perspective. It is unclear what your intentions are regarding the mortality data, for example. You need to be more explicit and pragmatic. My suggestion is to display the p-values when making comparisons in the table and to include figures showing the correlations with both p-values and correlation coefficients (r) between the variables.

Answer: Statistical descriptions and additional tables have been provided.

Harris Hip Score (mHHS): It would be beneficial for the lay reader to understand the Harris Hip Score (mHHS). A brief explanation of its interpretation would be helpful. For instance, scores range from 0 to 100, with higher scores indicating better function and less pain. A score above 70 is often considered satisfactory, while scores below 60 suggest poor outcomes. Providing this context would enhance the reader's comprehension.

Answer: Comment added on the materials & Methods section.

Clarification in Tables:

  • In Table 7, please clarify that the numbers 49 and 18 represent the 67 surviving patients at the 6-month follow-up.
  • In Table 8, please clarify that the numbers 47 and 16 represent the 63 surviving patients at the 1-year follow-up.

Answer: Clarified.

Mean Value Interpretation: The mean values shown in Tables 7 and 8 represent the differences concerning the pre-fracture status, which is a good idea. However, it would be beneficial to include polarity indicators (positive or negative). For example, while Group B showed the greatest differences, it is unclear whether these represent increases or decreases compared to the pre-fracture status.

Answer: Noted.

Conclusion Focus: Your conclusion has become somewhat unfocused. It could be more tightly aligned with the objectives of the study.

Answer: Modified accordingly.

Reviewer 2 Report

Comments and Suggestions for Authors

The manuscript  entitled “The Impact of Rotational Malalignment Following Intramedulary Nailing for Intertrochanteric Fractures on Patients’ Functional Outcome: A Prospective Study. " is very interesting and easy to read.

My comments;

1.      How long after the injury were the patients operated on and was there a relationship between incorrect positioning of the screws and the time between the injury and the surgery? 2.      It would be advisable to supplement the material with a table containing age, gender, causes of injury , comorbidities, side of injury 3.      Table 2 - D-angle abbreviation must be explained below the table 4.      Table 5 - D-angle and mHHS abbreviation must be explained below the table 5.      Tables 7 and 8  mHHS abbreviation must be explained below the table

6.      Can the authors explain the cause of death 6 and 12 months after the surgery.

Author Response

Dear reviewer, 

Thank you for the valuable comments and the time spent reviewing our work. 

1) How long after the injury were the patients operated on and was there a relationship between incorrect positioning of the screws and the time between the injury and the surgery?

Answer: The operation was performed 24-48 hours after the injury. (added in Materials & Methods section). Time to operation did not affect the patient’s outcome. (added in the first paragraph of the results section).

2) It would be advisable to supplement the material with a table containing age, gender, causes of injury, comorbidities, side of injury 3.

Answer: A table regarding the gender cause and side of injury is already included in the results section (table 1). The mean age of our population was 80.6 years, ranging from 47 to 99 years, which is also stated in the first paragraph of the results section. Regarding the comorbidities, as stated in the second to last paragraph of our discussion section, unfortunately, we did not emphasize on their possible relation, which may have limited the value of our results on patient mortality.

3) Table 2 - D-angle abbreviation must be explained below the table 4. 

Answer: The abbreviation was already explained in the paragraph above table 2. "The mean value of the femoral anteversion difference between the operated and healthy hip (D angle)". We removed the abbreviation from table 2, so that it does not cause any misunderstanding.

4) Table 5 - D-angle and mHHS abbreviation must be explained below the table 5. 

Answer: An explanation for D-angle and mHHS abbreviations was added below table 5.

5) Tables 7 and 8  mHHS abbreviation must be explained below the table

Answer: An explanation for mHHS abbreviation was added below tables 6, 7 and 8.

6) Can the authors explain the cause of death 6 and 12 months after the surgery.

Answer: We thought that presenting the exact cause of death was out of the scope of this study, especially due to the lack of a correlation with each patient's comorbidities. Nevertheless, we added a comment regarding the most common system involved (cardiovascular) and the most common diagnosis (myocardial infarction), in the second to last paragraph of our discussion.

Reviewer 3 Report

Comments and Suggestions for Authors

In this prospective study, the authors analyzed the impact of rotational malalignment following intramedullary nailing for intertrochanteric fractures on patients' functional outcomes. A total of 74 patients with intertrochanteric fractures treated with intramedullary nailing were included. The patients were divided into two groups based on the D angle (femoral anteversion difference) being less than or more than 15 degrees. Postoperative follow-ups were conducted at 6 months and 1 year, with the data used for further analysis. The results showed that rotational malalignment occurred in 31.1% of cases, impacting early postoperative function, with most patients recovering by one year; however, younger patients may have lower tolerance for this issue. The study highlights the need for advanced intraoperative methods and refined functional assessment tools for elderly patients.

Overall, the manuscript is well-written. The conclusions are substantiated by the data, and the discussion addresses study limitations, related work, and the importance of the findings. The figures are clear and informative. I have no further suggestions for improvement.

Author Response

Dear reviewer, 

Thank you for the valuable comments and the time spent reviewing our work. We are glad you find our work worth publishing in its current form.

Round 2

Reviewer 2 Report

Comments and Suggestions for Authors

Dear authors, thank you for responding to my comments. The corrections made in the manuscript are sufficient.